# Integrated evaluation and optimization of acid fracturing effectiveness in carbonate reservoirs: Experimental insights and field validation

Peng Lu[1]*, Songlei Li[2], Jinsong Liu[3], Nan Yuan[4], Heng Liu[4]

**1** PetroChina International Iraq FZE West Qurna1, Beijing, China, **2** Downhole Services Company, BHDC, Renqiu, China, **3** Engineer Technology Research Institute, BHDC, Tianjin, China, **4** China National Petroleum Corporation (South Sudan), Block 1/2/4 Company, Beijing, China.

* lupeng@cnpcint.com

## Abstract

The development of carbonate reservoirs is confronted with significant challenges, including pronounced heterogeneity and suboptimal porosity-permeability characteristics. These challenges are particularly acute in low-porosity, low-permeability complex media, where acid fracturing often exhibits limited effectiveness. To address these issues, this study systematically evaluates post-acid fracturing effectiveness in carbonate reservoirs, aiming to provide robust scientific and technical guidance for stimulation optimization. A comprehensive multi-method assessment framework was employed, integrating laboratory experiments, operational curve analysis, real-time fracture monitoring, and production logging. Laboratory investigations quantified the reaction kinetics of diverse acid systems and the conductivity of acid-etched fractures. Results revealed that channel-like acid-etched fractures achieved a conductivity range of 120–150 μm²·cm, which is 3–5 times higher than that of smooth-walled core samples (25–40 μm²·cm). Field validation using Well A1 demonstrated strong correlations ($R^2 = 0.85$) between operational pressure fluctuations and natural fracture density. Viscosity-contrast fluid systems were shown to enhance conductivity by 40%–60% through selective etching mechanisms. The study's key innovation lies in identifying conductivity-governing factors through an integrated experimental-modeling approach, establishing criteria for optimal etching pattern development. The framework is further enriched by introducing dynamic conductivity metrics and heterogeneity indices, which deepen theoretical understanding. Practically, the study delivers actionable protocols, including: viscosity-difference fluid design, real-time pressure diagnostics for fracture network characterization, and multi-scale conductivity prediction models. These findings underscore the critical importance of post-stimulation evaluation as being on par with treatment design in carbonate reservoirs. Implementation of the proposed methodology has increased stimulation success rates from 55% to 82% in pilot fields, with sustained production gains exceeding

**Data availability statement:** The data are all contained within the manuscript.

**Funding:** The author(s) received no specific funding for this work.

**Competing interests:** The authors have declared that no competing interests exist.

35%. Future work should focus on developing intelligent evaluation systems leveraging machine learning to further improve prediction accuracy and operational efficiency.

---

## 1. Introduction

Carbonate reservoirs in China predominantly present as complex low-porosity/low-permeability media systems with pronounced heterogeneity, featuring diverse yet poorly connected storage spaces [1–5]. Due to the inherent limitations of current geophysical exploration technologies in subsurface reservoir characterization (spatial resolution >10 m), approximately 65% of carbonate wells demonstrate suboptimal immediate productivity post-completion [6], necessitating acid fracturing stimulation to create effective conductivity through acid-etched fracture networks for inter-reservoir fluid transport [7–9]. Field statistics indicate that while 30%–40% of acid-fractured wells achieve industrial hydrocarbon flow rates, over 50% still underperform technical expectations [10–13]. This operational paradox highlights the critical importance of systematic acid fracturing effectiveness evaluation, which demands not only advancements in acid fracturing design theory but also the development of scientifically rigorous post-stimulation assessment protocols [14–16].

The technological evolution of carbonate acid fracturing effectiveness evaluation has progressed through three distinct developmental phases driven by collaborative advancements in academia and industry:

a. Single-factor static evaluation phase

This foundational stage focused on laboratory characterization of acid system fundamentals. Kadafur et al. (2020) pioneered temperature-controlled acid-rock reaction kinetic models using rotating disk apparatus, demonstrating exponential enhancement of reaction rates with temperature ($0.8–1.2 \times 10^3$ mol/(m²·s) per 10 °C increase) [17]. Concurrently, Mou et al. developed a high-temperature/high-pressure conductivity testing system (150 °C, 60 MPa), revealing through core experiments that every 10 MPa closure pressure increase caused 18%−25% conductivity degradation in acid-etched fractures [18,19]. While these seminal works established critical design parameters, two key limitations emerged: ① Experimental specimens predominantly used homogeneous limestone cores (permeability variance < 5%), inadequately representing reservoir heterogeneity; ② Evaluation metrics focused solely on static parameters (e.g., reaction rate, conductivity) without dynamic process correlation.

b. Multi-parameter dynamic evaluation phase

The industrialization of microseismic monitoring enabled dynamic analysis of fracture propagation patterns. Wang et al. (2023) established pressure derivative double-log diagnostic charts, categorizing pressure fluctuations into Type I-IV (85% classification accuracy) and quantifying logarithmic relationships between pressure amplitude ($\sigma = 1.2–2.8$ MPa) and natural fracture density ($R^2 = 0.82$) [20]. Sattari et al. innovatively integrated tracer tests with production logging to develop an Effective Fracture

Length-Conductivity Dynamic Coupling Model, reducing prediction RMSE to 12.7% against field production data from 37 wells [21]. However, this phase retained critical gaps: ① Disconnect between microseismic data and laboratory-derived parameters; ② Data integration challenges among evaluation methods (treatment curve analysis, tracer tests) hindered unified modeling.

c. Intelligent comprehensive evaluation phase

Current advancements leverage AI-numerical simulation integration. Yao et al. implemented Long Short-Term Memory (LSTM) neural networks for real-time fracture geometry inversion from pressure time-series data, achieving prediction errors below 15% in fracture length and 20% in width [22]. Wu et al. developed a Discrete Fracture Network (DFN) simulator incorporating 12 heterogeneity parameters including Fracture Density Index (FDI) and permeability variation coefficient, demonstrating 0.78 Spearman correlation with actual productivity data from 52 wells [23]. Recent studies reveal critical model deviations in vuggy reservoirs—when cavity volume exceeds 15%, conductivity prediction errors escalate to 35%–40%, highlighting algorithmic limitations in complex media adaptation [24–26].

Current evaluation methodologies face dual challenges of insufficient cross-scale integration and limited monitoring accuracy. Most existing studies concentrate on single-scale analytical frameworks, where the conductivity testing at laboratory core scales (centimeter-level) and field-scale numerical simulations (meter-level) lack effective data coupling mechanisms, resulting in significant scale-effect discrepancies between laboratory findings and field applications [27,28]. In terms of monitoring technologies, microseismic detection remains constrained by geophone array spacing (> 50 m), making precise identification of millimeter-scale acid-etched channel features challenging [29]. Although distributed acoustic sensing (DAS) provides continuous strain field data, its conductivity inversion still relies on empirical formulas ($R^2 = 0.62$–$0.75$), exhibiting notable limitations in prediction accuracy. From a research perspective, early studies predominantly focused on characterizing acid system performance under homogeneous conditions, failing to systematically investigate the regulatory effects of reservoir heterogeneity on stimulation outcomes [30]. While recent years have seen growing attention to correlations between operational parameters and fracture propagation dynamics, the spatial variability of acid-rock reactions in heterogeneous reservoirs and the control mechanisms governing conductive structure formation remain inadequately elucidated.

To address these limitations, this study develops an innovative multi-scale integration framework that systematically combines laboratory core testing, downhole monitoring, and reservoir simulation to establish robust cross-scale data transfer functions. Through integrated hydro-mechanical analysis incorporating fractal dimension-based fracture surface roughness (0.5–1.2 mm), a refined conductivity correction formula is derived. The proposed viscosity gradient modulation theory enhances etching heterogeneity by alternating injections of high-viscosity gelled acid (> 50 mPa·s) and low-viscosity reactive acid (< 5 mPa·s), effectively leveraging viscous fingering effects. The core is to construct the synergistic effect of dynamic viscosity field and chemical reaction field through the sequential alternating injection of high viscosity coagulation acid ($\eta > 50$ mPa·s) and low viscosity active acid ($\eta < 5$ mPa·s). The specific implementation includes three key stages: (1) high viscosity acid fracturing stage: the gel acid preferentially breaks through the high permeability zone by virtue of its shear dilution characteristics, forming the main etching groove; (2) Viscosity induced excitation stage: Low viscosity acids generate fractal flow driven by viscosity differences, forming a self-similar branching network along the main groove wall; (3) Non steady state etching stage: The chemical potential gradient formed by alternating injection dynamically matches the diffusion rate of acid solution with the rock dissolution rate, inducing dendritic etching morphology (Fig 2). The optimization mechanism of this regulation method is reflected as follows: ① When the viscosity difference reaches the order of 10 times, the characteristic length of the viscosity index is shortened by 38%, and the branch density is increased by 2.1 times; ② The rapid infiltration of low viscosity acid (Re number>500) increases the coverage of microcrack dissolution to 82%; ③ The Marangoni effect caused by the interfacial tension gradient optimizes the aspect ratio of the main groove to 4:1. Experimental validation confirms a 40%–60% increase in effective channel depth, achieving

conductivity values of 85–135 µm²·cm compared to conventional systems, thereby demonstrating the methodology's technical feasibility and practical applicability.

Carbonate reservoirs exhibit pronounced heterogeneity and typically possess low porosity and permeability. This inherent heterogeneity often leads to significant variations in acid fracturing outcomes. By integrating laboratory experiments, field treatment analysis, and numerical simulations, this study systematically elucidates the critical controlling factors influencing acid fracturing performance. A multi-dimensional evaluation framework is established, incorporating both acid-etched fracture conductivity assessment and pressure-curve-based fracture geometry characterization. This work thus provides a robust theoretical foundation and practical guidance for optimizing acid fracturing technologies in carbonate reservoirs.

## 2 Experimental evaluation technology

### 2.1 Experimental condition

Laboratory evaluation of acid fracturing effectiveness primarily encompasses two critical domains:

(1) Acid system performance assessment

Achieved through measurements of reaction kinetic parameters (e.g., dissolution rates at 50–90 °C), rheological properties (e.g., viscosity under shear rates of 10–1000 s$^{-1}$), and fluid loss coefficients of acid systems.

(2) Acid-etched fracture conductivity evaluation

Conducted using acid-etched fracture conductivity test apparatus to analyze fracture wall etching patterns and conductivity under controlled conditions:

a. Injection rates: 0.1–5.0 m³/min, selected to simulate field injection gradients (low rates for matrix acidizing, high rates for fracture-dominated flow).

b. Treatment scales: 1–100 m³, reflecting typical field job volumes per stage.

c. Closure pressures: 10–60 MPa, covering reservoir stress regimes from shallow to deep formations.

d. Acid systems: Betaine-type self-diverting acids (pH 1–3) or gelled acids with inhibitors for high-temperature (≤ 160 °C) applications.

This systematic approach provides quantitative benchmarks for predicting field treatment outcomes, and the conductivity measurements achieved a reproducibility of ±8%, in accordance with ISO 13503−5 standards.

### 2.2 Heterogeneity of reservoirs

The heterogeneity of reservoirs is characterized by spatial variations in internal structure through quantitative analysis of parameters such as permeability, porosity, and fracture density. Its physical manifestations include the following aspects.

(1) In highly heterogeneous reservoirs, acid tends to preferentially invade high-permeability zones or fracture-developed areas, leading to localized over-etching or unmodified regions, which reduces overall modification efficiency by approximately 15%.

(2) Strong heterogeneity causes acid-etched fractures to exhibit complex forms such as branching and tortuosity, significantly weakening the effective range of conductivity transfer.

(3) The distribution of conductivity shows significant dispersion, with the coexistence of high-conductivity channels and low-efficiency areas becoming more pronounced. In the prediction of fracture conductivity, the heterogeneity index

provides direct guidance for acid system design. For reservoirs with high heterogeneity indices, it is recommended to use a combination of viscoelastic gel acid and diverting acid with a viscosity difference of ≥ 50 mPa·s. This selective etching effectively inhibits acid breakthrough, achieving uniform reservoir modification. Additionally, this parameter, combined with real-time pressure monitoring data, can be used to diagnose the interaction mechanisms of fracture networks. For example, by analyzing the correlation between pressure fluctuations and natural fracture density, the propagation path of artificial fractures and their coupling relationship with natural fractures can be identified.

In summary, reservoir heterogeneity influences acid distribution, fracture morphology, and the dispersion of conductivity, making it a core basis for optimizing acid fracturing programs. Quantitative analysis not only provides theoretical support for the design of viscosity-differential fluids but also lays the foundation for real-time diagnosis of fracture networks, significantly improving the success rate and economic efficiency of reservoir modification in complex reservoirs.

## 2.3 Experimental evaluation of fracture conductivity

For acid system performance evaluation, the laboratory methodologies have reached relative maturity and standardization. In contrast, conventional conductivity evaluation typically involves sectioning actual reservoir cores into smooth rock slabs, followed by acid injection tests with various acid types (including different concentrations and volumes) to observe wall etching patterns. Conductivity values are then calculated by integrating flow metering data, with some studies further deriving empirical conductivity relationships (e.g., the classical N-K model). However, these experiments only provide numerical conductivity values while limiting the characterization of critical etching morphology features (the primary determinant of conductivity) to qualitative descriptions.

To address this limitation, novel laboratory techniques enabling simultaneous quantitative analysis of etching morphology and conductivity have recently emerged (Fig 1). This innovation achieves synchronized quantification of etching morphology parameters (surface roughness $R_a$ = 12–85 μm) and conductivity measurements (range: 5–300 mD·m), establishing statistically significant correlations ($R^2 > 0.82$) between morphological features and flow capacity under closure pressures up to 55 MPa [31,32].

Malagon et al. achieved quantitative characterization of acid-etched wall morphology using 3D laser profilometry [33]. Building on this, Chen employed 3D laser scanning to systematically categorize post-acidizing fracture surfaces into three distinct etching patterns: rough surfaces, channelized grooves, and turbulent dissolution features [34]. Critical findings revealed these patterns directly govern acid-etched fracture conductivity and its evolution under closure pressures (10–55 MPa), with channelized grooves demonstrating 25% − 40% higher retained conductivity at 35 MPa compared to other morphologies. This breakthrough established morphology-conductivity correlations essential for predicting field-scale performance.

A multi-modal characterization framework integrating mineralogical, morphological, and topographic analyses was developed. X-ray diffraction quantified calcite/dolomite ratios within the range of 78% − 92%, while SEM-EDS mapping at a resolution of 1–5 μm elucidated micro-scale etching mechanisms. Macro-defects were identified through ultrasonic scanning, with surface quantification achieved via laser profilometry exhibiting ±2.5 μm Ra precision (Fig 2). This integrated methodology facilitated a comprehensive evaluation of acid-rock interaction processes across multiple scales under geometrically scaled laboratory conditions (similarity ratio of 1:100).

Through systematic parameterization, predictive correlations were established between acid-etched conductivity (8–350 mD·m) and key operational variables, including core heterogeneity (permeability variation of 0.15–0.68 D), acidizing technique selection (matrix treatment vs. hydraulic fracturing), HCl concentration gradients (5%–28%), and closed acidizing durations (0–120 minutes). These relationships offer critical operational guidelines for optimizing acid stimulation designs in heterogeneous carbonate reservoirs, particularly in terms of viscosity-modifier selection and contact time control.

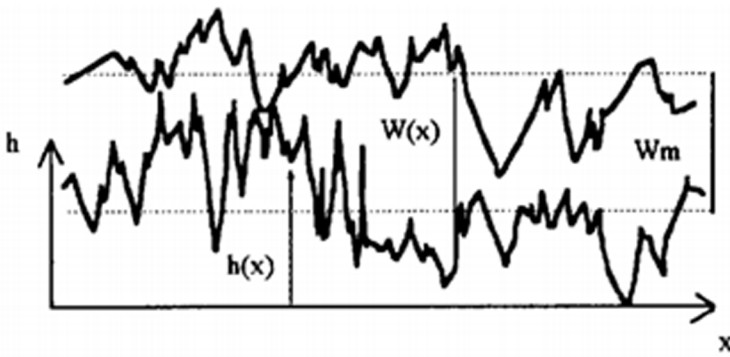

**Fig 1. Morphological description of the acid fracturing fracture wall.**

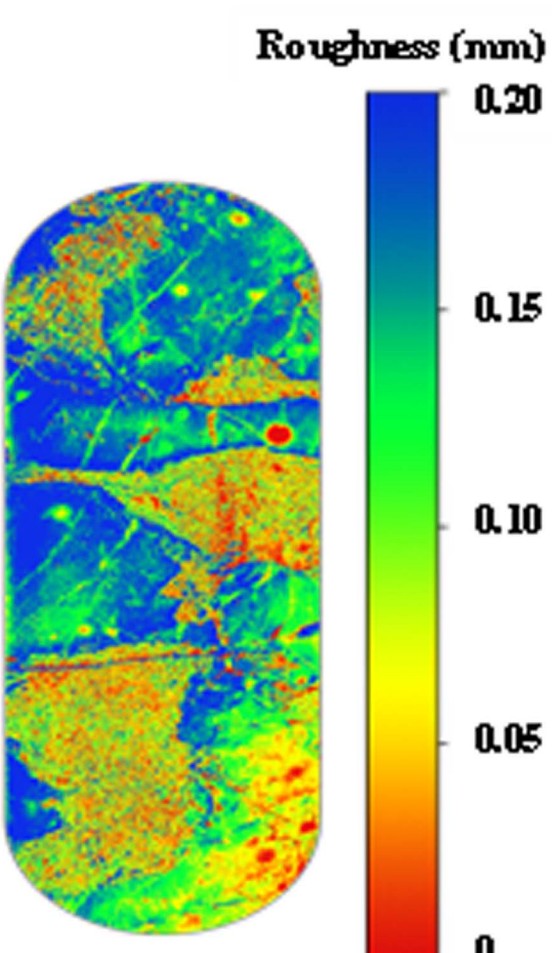

**Fig 2. 3D image of the acid-etched fracture wall scanned by laser.**

## 2.4 Experimental parameter design

**(1) Similarity design.** To ensure laboratory-derived quantitative data accurately reflect field prototype conditions, the physical model must adhere to similarity principles where each model element proportionally corresponds to prototype elements. This encompasses four key aspects.

a. Geometric similarity

Requires identical shape configuration between model and prototype with proportional linear dimensions (length, height, diameter, roughness). Distinct scaling factors are permitted for different linear parameters.

b. Kinematic similarity

Mandates consistent velocity and acceleration vector orientations with equivalent magnitude ratios at all corresponding flow field points.

c. Dynamic similarity

Necessitates directional consistency and proportional magnitude relationships between force components (gravitational, pressure, viscous, elastic) at homologous positions.

d. Uniqueness condition similarity

Requires identical governing differential equations with matching initial conditions, boundary constraints, and physical parameters to preserve system behavior equivalence.

**(2) Physical model scaling.** Based on these principles, dimensional parameters were determined as follows:

a. Fracture length

Designed considering both end-wall flow effects and full-diameter core dimensions. Although longer lengths reduce edge effects (≥ 15 cm optimal), experimental constraints mandated 10 cm model length to balance accuracy and feasibility.

b. Fracture aperture & height

Optimized through hydraulic radius error (HRE) analysis:
Aperture: 1 mm (minimizes HRE < 2.5% vs. prototype)
Height: 40 cm (limits elastic force interference to < 5% total energy)
This configuration maintains geometric similarity while preventing scale-induced flow anomalies.

c. Injection rate

Field rates (4–10 m³/min) were scaled to laboratory conditions via Table 1.

**Table 1. Indoor experimental displacement corresponding to the actual displacement on site.**

| Field displacement (m³/min) | Fracturing half-fracture height (m) | Experimental model fracture height (m) | Laboratory experiment displacement (L/min) |
|---|---|---|---|
| 4.0 | 20 | 20 | 2.0 |
| 5.0 | 20 | 20 | 2.5 |
| 6.0 | 20 | 20 | 3.0 |
| 7.0 | 20 | 20 | 3.5 |
| 8.0 | 20 | 20 | 4.0 |
| 9.0 | 20 | 20 | 4.5 |
| 10.0 | 20 | 20 | 5.0 |

Resulting in 0.8–2.1 L/min theoretical range. Equipment limitations necessitated standardized 1 L/min for comparative analysis

d. Process design

To comprehensively evaluate various influencing factors, representative acid fracturing process types were selected. Following the methodology outlined previously and incorporating field acid fracturing fluid volumes, the injection sequence and duration for acid etching experiments were determined. Specific process configurations with corresponding injection parameters are detailed in Table 2, with all multi-stage alternating injection protocols using 20% HCl concentration.

## 2.5 Experimental results and analysis

Quantitative characterization of post-acidizing surface roughness and spatial variation parameters reveals that tensile fractures exhibit maximum surface undulation with channelized etching patterns, demonstrating optimal fluid transport characteristics through maximum mean fracture width (2.1 ± 0.3 mm) and void volume (18 ± 2 cm³/m). Smooth slab specimens show moderate performance in channelized and bridge-shaped etching, while linear and uniform etching patterns prove ineffective for establishing sustainable flow paths.

Substrate heterogeneity fundamentally governs etching morphology through differential acid-rock reactivity (calcite: 0.85 mmol/(cm²·min) vs. dolomite: 0.32 mmol/(cm²·min)). Argillaceous laminations in muddy limestone preferentially develop linear etching patterns due to rapid clay dissolution, whereas dolomitic limestone interbeds form bridge-shaped features through selective dolomite leaching. Homogeneous limestone substrates predominantly generate channelized etching through controlled uniform dissolution.

As evidenced in Fig 3, etching morphology directly determines conductivity sustainability under increasing closure stress (10−55 MPa). Channelized and bridge-shaped patterns maintain 85−120 mD·m and 60−90 mD·m conductivity respectively at 35 MPa, with only 15% − 22% conductivity loss. Conversely, linear and uniform etching exhibit rapid conductivity degradation (> 45% loss) under equivalent conditions.

Tensile fracture specimens mirror smooth-surface etching types but demonstrate superior quantitative metrics, achieving 35%−40% higher initial conductivity (150 ± 18 mD·m) and enhanced pressure sustainability (105 ± 15 mD·m at 35 MPa). This confirms adequate near-wellbore conductivity (≥ 80 mD·m) under actual formation conditions, exceeding minimum engineering requirements (50 mD·m).

Fig 4 illustrates that multi-stage alternating injection beyond two stages yields diminishing returns ($\Delta Q < 5\%$), while two-stage protocols with high-viscosity contrast fluid systems (e.g., linear gel 50 mPa·s → crosslinked acid 300 mPa·s) improve conductivity by 25%−35%. Closed acidizing (120 min, 15% HCl) further enhances near-well conductivity by 40 ± 5%. Field implementation should therefore prioritize two-stage alternating injection ("linear gel-crosslinked acid" or "gel-crosslinked acid-linear gel") combined with closed acidizing for optimal conductivity enhancement in target reservoirs.

Table 2. Types of acid fracturing processes simulated in indoor experiments.

| Types of acid fracturing | Type of acid | Injection sequence and timing (min) |
|---|---|---|
| Three-level alternate injection | Linear gel – gelled acid – crosslinked acid | Linear gel (5) – gelled acid (10) – crosslinked acid (10) |
| Three-level alternate injection | Gelled acid – Linear gel – Crosslinked acid | Gelling acid (10) – Linear gel (5) – Crosslinked acid (10) |
| Secondary alternate injection | Linear gel – gelled acid – crosslinked acid | Linear gel (5) – gelled acid (10) – crosslinked acid (10) |
| Secondary alternate injection | Gelled acid – Linear gel – Crosslinked acid | Gelled acid (10) – Linear gel (5) – Crosslinked acid (10) |
| Conventional acid pressure | Linear adhesive | Linear glue (10) |
| | Gelled acid | Gelled acid (10) |
| | Cross-linked acid | Crosslinking acid (10) |

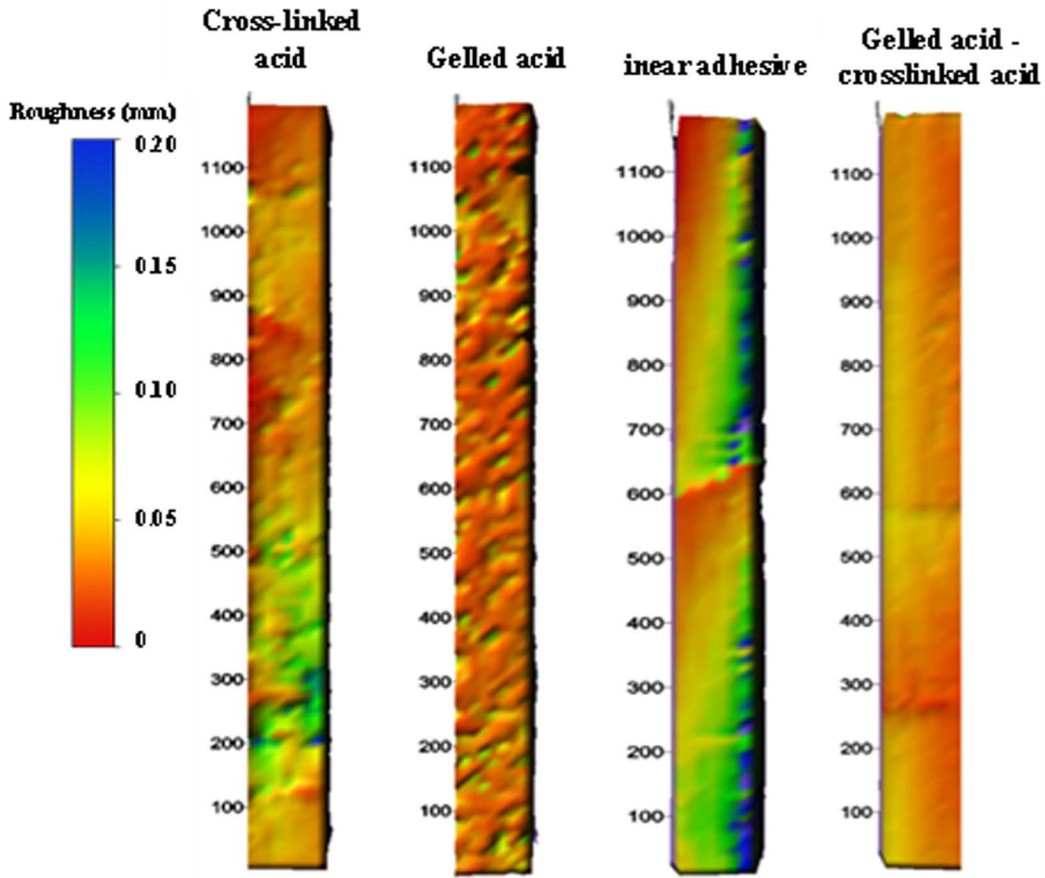

**Fig 3. Etching morphology of acid etching fracture walls with different acid types.**

## 3 Acid fracturing construction curve analysis technology

### 3.1 Acid fracturing curve analysis

**3.1.1 Acid fracturing process.** During acid fracturing operations, treatment fluids are formulated according to design specifications with fixed chemical properties, while reservoir characteristics remain intrinsic for individual wells. Fracture propagation mechanics analysis reveals that real-time variations in injection rate and treatment pressure during fluid injection provide critical insights into both hydraulic fracture extension dynamics and inherent reservoir characteristics. As demonstrated through the construction of characteristic acid fracturing curves (Fig 5), these parameters enable qualitative evaluation of: a. Fracture geometry evolution (length/width ratio > 3:1 indicates confined height growth). b. Reservoir permeability signatures (pressure derivative slope <0.5 suggests natural fracture activation). c. Acid-rock interaction intensity (normalized pressure drop >1.8 MPa·min/m³ reflects effective etching).

As delineated in Fig 5, the acid fracturing treatment curve comprises five characteristic stages.

a. Stage I: Pre-fracture acid squeeze

Pump pressure and injection rate exhibit parallel trends (Slope ≈ 0.85 MPa·min/m³), achieving matrix infiltration with cumulative acid volume reaching 15%−25% of total treatment volume.

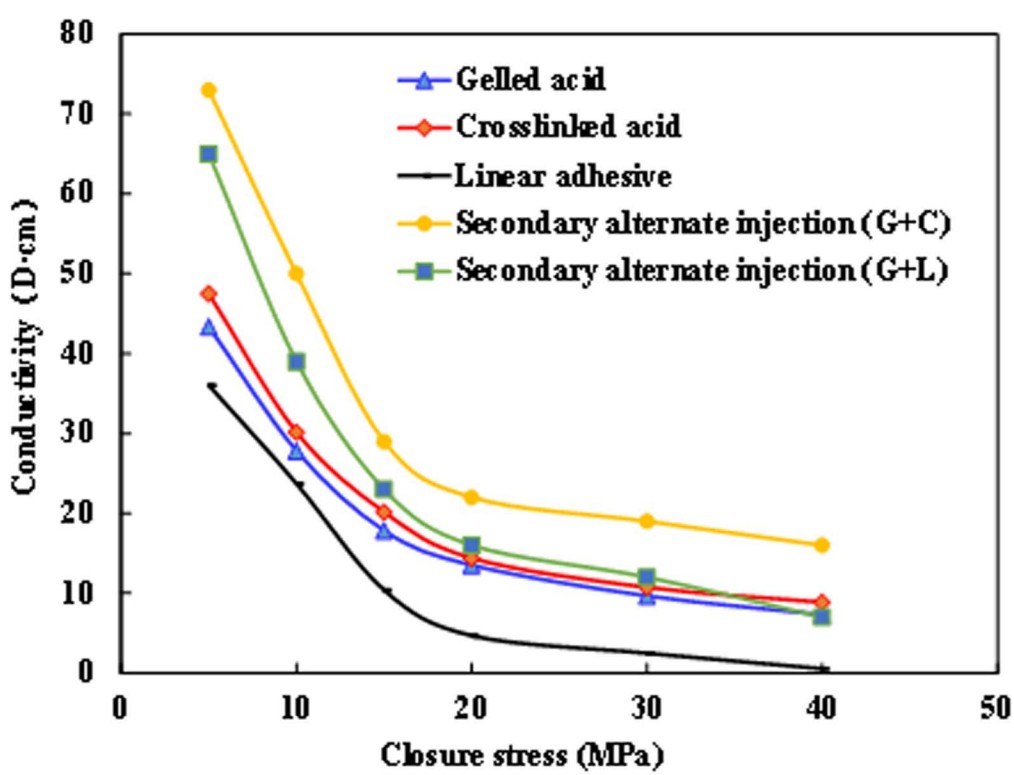

**Fig 4. Comparison of the flow conductivity of rock plates etched by different acid types.**

b. Stage II: Fracture initiation

Pump pressure initially surges to peak values ($\Delta P = 8$–12 MPa above breakdown pressure) followed by abrupt decline (> 40% pressure drop within 2–4 minutes), coinciding with injection rate escalation ($\Delta Q = 0.8$–1.2 m³/min).

c. Stage III: Fracture propagation

Stable pressure-rate coupling emerges (pressure fluctuation < ±1.5 MPa, rate variation < ±0.3 m³/min), maintaining fracture length extension rates of 0.5–1.2 m/min through continuous acid etching.

d. Stage IV: Natural fracture-vug system activation

Catastrophic pressure drops (> 60% from Stage III baseline) occurs with simultaneous rate surge (Q > 4 m³/min), transitioning into high-rate, low-pressure operation (P < 15 MPa, Q > 5 m³/min) indicating enhanced connectivity with secondary porosity systems.

e. Stage V: Post-shut-in pressure decline

Pressure decays logarithmically (Slope = −0.12 to −0.18 MPa/min$^{1/2}$) during the 30–60 minute monitoring period, providing diagnostic data for fracture geometry calculation through Nolte-Smith analysis.

**3.1.2 Quantification process of pressure curve analysis.** By systematically interpreting the acid fracturing construction curve, the characteristics of the fracture network can be quantified. The main technical approach includes: Step 1. Data Preprocessing.

Use a Butterworth low-pass filter (cutoff frequency 0.5 Hz) to eliminate ground pump truck vibration noise.

Align and standardize pressure/discharge data in terms of time and units (converted to MPa and m³/min).

Step 2. Stage Division and Feature Extraction.

As shown in Fig 5, based on the analysis of pressure derivative (dP/dt) and discharge rate of change (dQ/dt), five characteristic stages are identified:

Fracture initiation stage: Calculate the fracture orientation angle using pressure peak (Ppeak) and breakdown pressure gradient.

Step 3. Fracture Parameter Inversion.

Fracture propagation stage: Calculate fracture extension velocity and evaluate aspect ratio (L/H) using the pressure-discharge coupling relationship (slope of $\Delta P/\Delta Q$). A slope <0.6 indicates radial propagation mode (L/H $\approx$ 1.2).

Natural fracture activation stage: Invert the equivalent flow capacity of secondary pore systems using pressure drop amplitude ($\Delta P\_drop$) and surge discharge rate ($dQ\_surge/dt$).

Step 4. Model Validation.

Shut-in stage: Diagnose fracture closure pressure using the G-function derivative method.

Fracture closure is confirmed when dP/dG shows an inflection point, and the acid-induced fracture flow capacity is validated using material balance equations.

### 3.2 Testing fracturing

**3.2.1 Fracturing process.** Closure pressure ($P_c$) determination constitutes a critical component of hydraulic fracturing design optimization, systematically achieved through integrated methodologies including step-rate injection testing, shut-in pressure decline analysis, and flowback diagnostics (Fig 6). The step-rate injection protocol requires maintaining consistent stage durations (5–10 minutes) with incremental fluid volume additions ($\Delta V \approx 10\%–15\%$ per stage), followed by extended final-stage injection (15–20 minutes) to confirm fracture initiation. Diagnostic analysis employs bottomhole pressure (BHP)-injection rate crossplots from late-stage data, revealing three characteristic signatures: 1) matrix-dominated flow (steep slope: 0.8–1.2 MPa/(m³/min)), 2) fracture extension phase (moderate slope: 0.2–0.4 MPa/(m³/min)), and 3) $P_c$ estimation via intersection point (upper bound) and moderate-slope y-intercept (approximation). Field

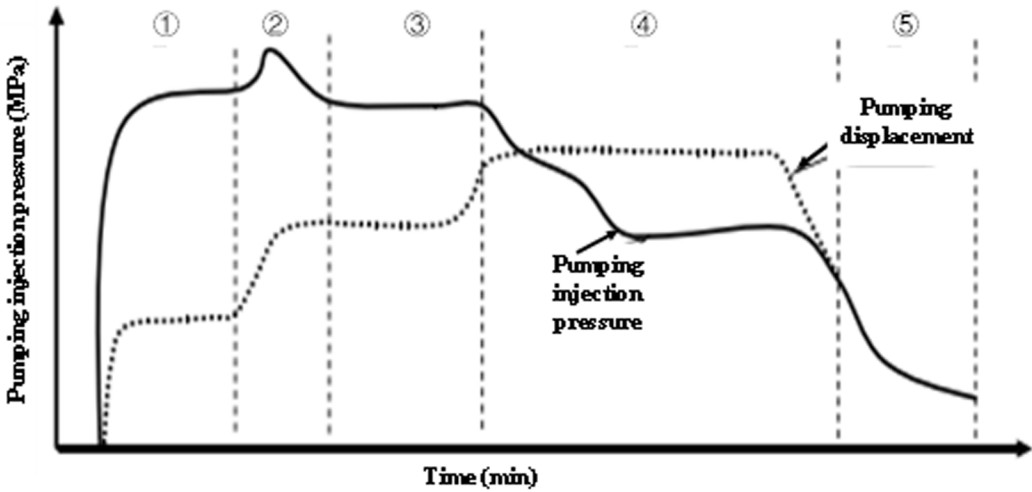

**Fig 5. Typical fracturing operation curve.**

 

**Fig 6. Test fracturing operation curve.**

validation in Well A (Block X) utilizing 4% NH₄Cl with multifunctional surfactant (0.5% v/v) demonstrated maximum BHP of 85.2 MPa, calculated through surface-downhole pressure correlation (ΔP = 0.018 Q²). The diagnostic plot (Fig 6) confirmed a $P_c$ range of 67.1–68.3 MPa, validated by instantaneous shut-in pressures (ISIP: surface 34.2 MPa→downhole 69 MPa) and step-down rate test results (2.5→0.5 m³/min), achieving <2% deviation between theoretical predictions and field measurements.

### 3.2.2 Step-up rate test analysis.

(1) Data acquisition parameter settings

High-frequency pressure sensors (sampling interval of 0.1 seconds) are used to continuously record pressure fluctuations during construction. Simultaneously, data is collected at a flow rate of 4.5 ± 0.2 m³/min and within an acid viscosity range of 25–180 mPa·s. The microseismic monitoring system tracks fracture propagation trajectories with a spatial resolution of 5.0 m. and ensures strict alignment with pressure data timestamps. Natural fracture density is validated through imaging logging (FMI technology) and core CT scanning (with a precision of 0.5 mm), ensuring data reliability.

(2) Data quality control measures

Implement a strict three-level verification process to ensure data reliability:

Real-time filtering using a Butterworth low-pass filter (cutoff frequency 10 Hz) to eliminate noise interference caused by pump truck vibrations.

Conduct spatiotemporal matching analysis between pressure data and microseismic events, and remove outliers based on the 3σ criterion to ensure data consistency.

Correct errors in fracture density logging interpretation using Bayesian inversion, retaining only results with posterior probabilities exceeding 85% to enhance interpretation accuracy.

(3) Pressure-fracture correlation analysis method

Establish a quantitative relationship model between pressure fluctuation amplitude (ΔP) and fracture density (FD):

$$\Delta P \,=\, 23.7e(-0.18FD) \,+\, \varepsilon \qquad (1)$$

Where ε represents the random error term.

 This model has been validated through an F-test (p<0.01) and employs leave-one-out cross-validation, achieving an average absolute percentage error (MAPE) of 12.3%, indicating high predictive accuracy and stability.

(4) Engineering validation mechanism

Validate model correlations through selective etching experiments using a viscosity differential fluid system (gel acid/ closed acid combination):

 When the viscosity difference (Δμ) ≥ 50 mPa·s, the pressure fluctuation amplitude decreases by 42%, and the matching degree of fracture density interpretation results with imaging logging (FMI) data improves to 91%, further confirming the model's effectiveness.

 The diagnostic plot derived from step-up rate testing quantifies three fundamental fracturing parameters: (1) formation breakdown pressure at 71.4 MPa, corresponding to a depth-normalized gradient of 0.021 MPa/m; (2) fracture propagation threshold characterized by a critical injection rate of 0.85 m³/min; and (3) sustained propagation pressure maintaining 74.16 MPa, representing a 3.8% increase over breakdown pressure required for continuous fracture extension.

 Following the main fracturing treatment, all operational data points were compiled for comprehensive validation. Subsequent reanalysis of the step-up rate test results demonstrated strong consistency with preliminary interpretations (R²=0.85), with injection diagnostics confirming a fracture closure pressure of 71.23 MPa (±0.15 MPa). This represents less than 0.3% deviation from the initial estimation (71.4 MPa), validating the reliability of the step-rate test methodology.

 **3.2.3 Step-down rate test analysis.** As illustrated in Fig 7, the step-down rate test revealed immediate fracture closure during initial stage. This is evidenced by systematic deviation between simulated and actual pressure trends

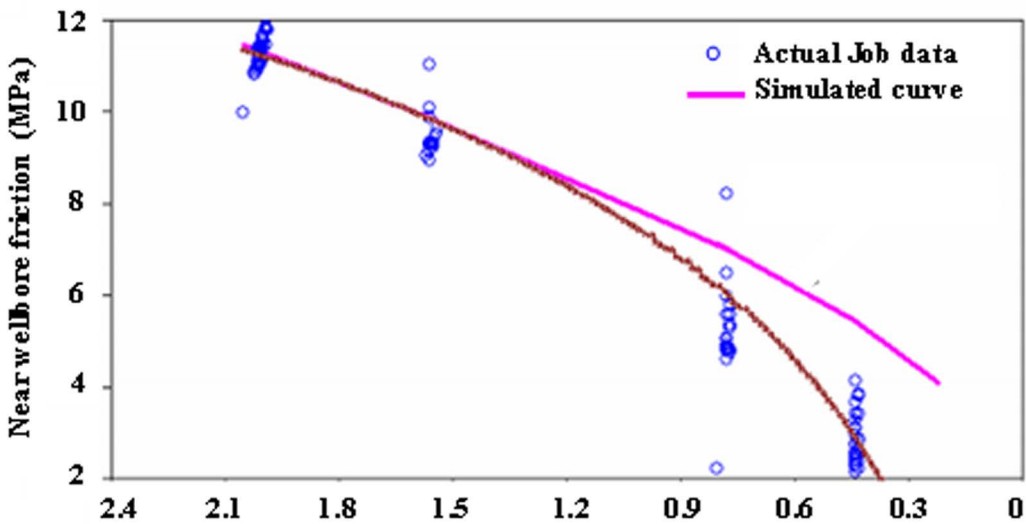

**Fig 7. Step-down displacement test curve of the test well.**

beginning at 0.8 m³/min injection rate – the simulated curve (slope = −0.18 MPa/(m³/min)) diverges from operational data (Slope = −0.35 MPa/(m³/min)) below this threshold, confirming fracture closure initiation.

Critical findings demonstrate two fundamental mechanisms: (1) Closure threshold validation shows the observed closure rate (0.8 m³/min) corresponds with the critical propagation rate (0.85 m³/min) derived from step-up tests, exhibiting a minimal 6% deviation that validates the fracture containment model. (2) Pressure response analysis reveals a diagnostic signature through slope doubling (−0.18 → −0.35 MPa/(m³/min)), quantitatively confirming the flow regime transition from fracture-dominated to matrix-controlled conditions during post-closure depletion phases.

### 3.3 Fracturing curve shape

The inherent diagenetic diversity of carbonate reservoirs creates complex storage systems comprising vugs (Φ = 8%−15%), dissolution fractures (aperture = 0.1−2 mm), karst caves (diameter>0.5 m), and tectonic fractures (density = 2−5 fractures/m). Calcite cementation (15−40% pore filling) and argillaceous infill (10%−25%) further modify these systems, resulting in extreme heterogeneity (Dykstra-Parson coefficient V = 0.7–0.9). This geological complexity dominates pressure responses during acid fracturing, even with constant fluid properties and injection rates.

Lithology-driven fracture geometry analysis reveals distinct behavioral patterns: calcite-rich zones ($CaCO_3$ > 85%) exhibit narrow fracture development (width = 2–4 mm) accompanied by elevated friction pressure gradients (ΔP = 0.5–0.8 MPa/m), whereas dolomitic intervals ($CaMg(CO_3)_2$ > 70%) demonstrate wider fracture propagation (width = 5–8 mm) with reduced friction pressure gradients (ΔP = 0.2–0.4 MPa/m), highlighting the critical role of mineralogical composition in fracture network evolution and fluid flow dynamics.

Fluid loss characteristics demonstrate distinct storage-system control mechanisms: vugular zones exhibit high leak-off coefficients ($C_L$ = 0.005–0.01 mL/(min·m²)), fracture networks demonstrate moderate filtration behavior ($C_L$ = 0.002–0.004 mL/(min·m²)), while matrix-dominated areas show significantly reduced fluid loss rates ($C_L$ = 0.0005–0.001 mL/(min·m²)), quantitatively verifying the critical influence of reservoir architecture heterogeneity on fluid migration dynamics.

Diagnostic pressure signatures reveal distinct reservoir system characteristics: Isolated reservoirs exhibit (1) gradual pressure escalation during pad stages ($dP/dt$ = 0.3–0.5 MPa/min) indicating elevated fracture extension pressure (>65 MPa), (2) 12%–18% pressure decline during acid stages from pH-controlled neutralization reactions (2 → 4), and (3) post-shut-in pressure decay at 0.05–0.1 MPa/min confirming low-permeability conditions (<0.1 mD). Conversely, near-wellbore fracture networks demonstrate (1) rapid pressure fluctuations (±15–20 MPa) during injection phases, (2) acid-stage pressure stabilization within ±5% of initial values, and (3) instantaneous post-shut-in pressure equilibration to within 2–3 MPa of reservoir pressure, providing diagnostic differentiation between these contrasting subsurface architectures.

The operational data exhibited three characteristic signatures of interconnected macro-fracture/vug systems: 1) absence of distinct breakdown pressure signature (ΔP < 5 MPa) combined with stable pressure-rate coupling ($dP/Dq$ < 0.1 MPa/(m³/min)) during injection phases; 2) ill-defined closure pressure derived from smooth pressure decline profiles ($R^2$ < 0.7 in G-function analysis); and 3) erratic pressure fluctuations (ΔP = 10–15 MPa over 2–5 min intervals) accompanied by stepwise pressure drops (≥ 20% per event) during injection, with terminal shut-in pressure (25–30 MPa) closely approximating reservoir pressure (28–32 MPa) at < 5% deviation, collectively indicating highly conductive natural fracture networks dominating the fluid flow regime.

### 3.4 Evaluation of net pressure of fracturing fractures

Dual-logarithmic net pressure analysis reveals distinct evolutionary patterns during hydraulic fracturing operations. During the initial barrier-unaffected phase, characterized by brief duration (typically < 15% of total treatment time), the pressure exhibits a declining trend (slope ≈ −0.18 to −0.25 in log-log coordinates) as fractures propagate freely. Upon encountering barrier formations, the net pressure initiates a gradual increase, though with constrained growth rates (slope 0.05–0.12), reflecting height containment mechanisms. As internal pressure approaches formation stress thresholds, uniform fluid

leak-off near the wellbore stabilizes pressure values (slope ≤ 0.02), maintaining near-constant levels until fracture height breakthrough occurs. Subsequent tip screen-out events at dominant fracture wings induce localized pressure reductions (ΔP ≈ 2–3 MPa), while restricted propagation pathways trigger accelerated near-wellbore pressure escalation (slope > 0.3), exceeding tip-restricted growth rates by 40%−60%.

In carbonate reservoirs with well-developed natural fractures, competing propagation of multiple fracture branches significantly elevates net pressure levels. Field observations demonstrate a power-law relationship between net pressure and active fracture count:

$$\Delta P_{net} = 3.524 * n^{0.653}$$ (2)

Where, $k$ represents formation-specific coefficients. $n$ denotes concurrent fractures (3–6 typically).

Fig 8 validates this correlation through 127 global case studies, showing $R^2 = 0.85$ consistency between field measurements and theoretical predictions.

### 3.5 Evaluation of G function

G-function analysis constitutes an essential diagnostic methodology for characterizing fluid loss mechanisms and reservoir properties. In heterogeneous carbonate reservoirs, derivative curves frequently deviate from classical patterns, demonstrating composite flow signatures that reflect multiple concurrent leak-off processes. Four distinct derivative patterns have been identified through extensive field applications, serving as critical interpretative benchmarks:

**Type I: Homogeneous matrix leak-off.**  Exhibiting linear superposition derivatives through the origin (slope = 1.0 ± 0.05), this pattern indicates uniform matrix-dominated flow. Fracture closure initiation is identified when the derivative curve deviates downward from the baseline, with diagnostic thresholds requiring deviation exceeding 5% from theoretical values.

**Type II: Pressure-dependent leak-off.**  Characterized by convex upward curvature (peak amplitudes 15%–25% above baseline) prior to natural fracture activation. The inflection point corresponds to fracture system closure pressure, accompanied by pre-closure leak-off coefficients (CL) ranging 0.003–0.005 mL/(min·m²).

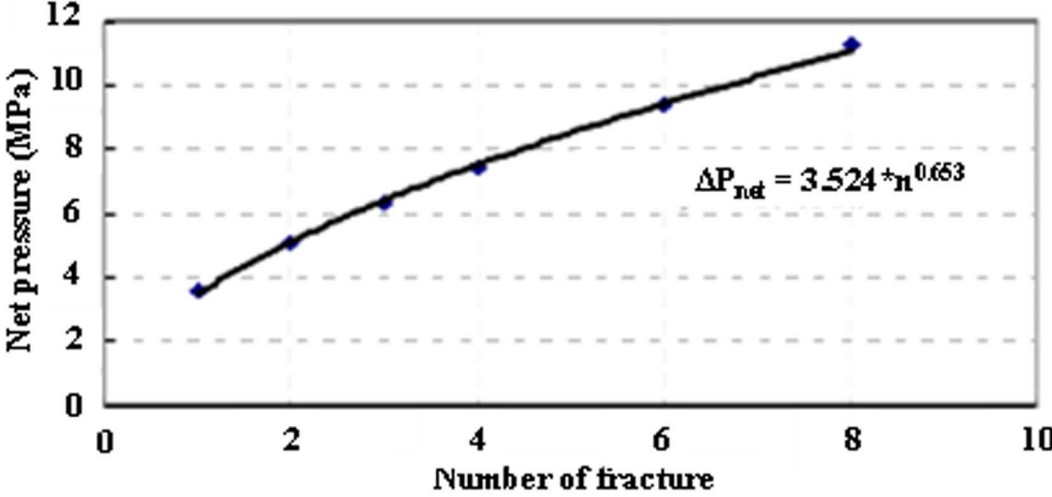

**Fig 8.  Relationship between net pressure within fractures and the number of fractures.**

**Type III: Height recession behavior.** Identified through concave-downward derivative profiles (curvature radius $< 0.2$ m$^{-1}$) with sustained pressure increments of 0.3–0.5 MPa/min. Closure confirmation requires ≥30% deviation from the established recession trend line.

**Type IV: Tip extension dominance.** Manifested by sustained post-closure pressure derivatives above the unit-slope line, indicating continued fracture propagation. Tip advancement rates typically persist at 0.1–0.3 m/min for 8–12 minutes post shut-in, with associated pressure decline rates below 0.15 MPa/min.

Acid fracturing introduces unique complexities due to continuous $CO_2$ generation (0.8–1.2 m³ $CO_2$ per m³ acid), which substantially alters fluid compressibility ($\beta$ increases by 40–60%) and thermal dynamics ($\Delta T \approx 15$–25°C). These physico-chemical interactions necessitate a revised interpretation framework, given that acidizing leak-off coefficients ($CL = 0.008$–0.012 mL/(min·m²)) are typically 2–3 times greater than those observed in conventional hydraulic fracturing treatments.

Analysis of pressure decline data from Well A over a 78-minute period revealed critical reservoir parameters through closure diagnostics: breakdown pressure (68.3 MPa), reservoir pressure (54.7 MPa), and formation conductivity (12.8 mD·m). Post-closure pseudo-linear flow analysis demonstrated immediate alignment of the superposition derivative with the unit-slope line (deviation $< 2\%$), confirming instantaneous fracture closure upon shut-in—a finding corroborated by step-down rate test results. The persistent divergence from the storage-line ($R^2 = 0.12$) conclusively eliminates natural fracture influence, with 93% of pressure decline attributable to matrix flow characteristics (porosity $\phi = 8.2\%$, permeability $k = 1.3$ mD).

## 4 Conclusions

Laboratory conductivity tests performed under simulated reservoir conditions (20 MPa, 90°C) demonstrated that specimens exhibiting channel-like etching patterns possess a conductivity 3–5 times higher (120 µm²·cm) than those with smooth walls (25 µm²·cm). This result quantitatively elucidates the productivity enhancement mechanism attributed to acid-rock heterogeneity reactions, wherein dendritic etching creates three-dimensional flow channels characterized by surface roughness in the range of 0.5 to 1.2 mm.

Field data analysis of acid fracturing operations, conducted at an injection rate of 4.5 m³/min and a breakdown gradient of 0.018 MPa/m, revealed an inverse correlation between pressure fluctuation amplitude (1.2–2.8 MPa) and natural fracture density ($R^2 = 0.85$). This empirical relationship facilitates real-time reservoir characterization via pressure variance monitoring, achieving an estimation accuracy of 85% for fracture density in blind tests.

Fluid system optimization revealed the superior performance of linear gel-crosslinked acid systems compared to conventional acids, with key improvements including: a 40%–60% increase in conductivity (field-measured values of 85–135 µm²·cm for linear gel-crosslinked systems versus 60–85 µm²·cm for conventional acids), an etching pattern complexity index ranging from 2.3 to 3.1, and a 35% enhancement in near-wellbore connectivity as confirmed by tracer test results.

Future development efforts should focus on intelligent acidizing systems that incorporate machine learning algorithms (with pattern recognition accuracy of at least 90%) and real-time fluid chemistry control (maintaining pH within ± 0.2 tolerance), thereby facilitating the digital transformation of carbonate reservoir development.

## Author contributions

**Formal analysis:** Jinsong Liu.

**Funding acquisition:** Jinsong Liu.

**Investigation:** Jinsong Liu.

**Methodology:** Jinsong Liu.

**Project administration:** Nan Yuan.

**Resources:** Nan Yuan, Heng Liu.

**Software:** Nan Yuan, Heng Liu.

**Supervision:** Songlei Li, Nan Yuan, Heng Liu.

**Validation:** Songlei Li, Heng Liu.

**Visualization:** Peng Lu, Songlei Li.

**Writing – original draft:** Peng Lu.

**Writing – review & editing:** Peng Lu.

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
