## [Decision Letter · Decision Letter 0]

Dear Dr. Lu,

Thank you for submitting your manuscript to PLOS ONE. After careful consideration, we feel that it has merit but does not fully meet PLOS ONE’s publication criteria as it currently stands. Therefore, we invite you to submit a revised version of the manuscript that addresses the points raised during the review process.

We look forward to receiving your revised manuscript.

Kind regards,

Karthik Kannan, Ph. D.,

Academic Editor

PLOS ONE

Journal Requirements:

4. We note that your Data Availability Statement is currently as follows: Data is provided within the manuscript or supplementary information files.

Reviewers' comments:

Reviewer's Responses to Questions

**Comments to the Author**

1. Is the manuscript technically sound, and do the data support the conclusions?

Reviewer #1: Partly

Reviewer #2: Yes

2. Has the statistical analysis been performed appropriately and rigorously?

Reviewer #1: No

Reviewer #2: Yes

3. Have the authors made all data underlying the findings in their manuscript fully available?

Reviewer #1: No

Reviewer #2: Yes

4. Is the manuscript presented in an intelligible fashion and written in standard English?

Reviewer #1: Yes

Reviewer #2: Yes

Reviewer #1: Present paper describes a study that aims to “Integrated evaluation and optimization of acid fracturing effectiveness in carbonate reservoirs: experimental insights and field validation ”. While the study's topic is of relevance to this journal, the manuscript's content requires major revisions to improve its scientific quality. Upon careful review, I have identified several issues that need to be addressed before the paper can be accepted for publication.

1. Clarify the Concept of "Viscosity Gradient Modulation": In the abstract (Page 10), the phrase "alternating injections of high-viscosity gelled acid (>50 mPa·s) with low-viscosity reactive acid (<5 mPa·s)" is introduced but lacks clarity. Please elaborate on the procedure, the rationale behind alternating viscosities, and how this modulation specifically enhances etching heterogeneity, ideally with a simplified explanation for clarity.

2. Simplify Technical Language in the Introduction: The sentence "The development of carbonate reservoirs is constrained by strong heterogeneity and low porosity-permeability characteristics, particularly in complex media reservoirs where acid fracturing effectiveness exhibits significant variability" (Page 8) is overly complex. Please break it into shorter, clearer sentences to improve readability, e.g., "Carbonate reservoirs are highly heterogeneous and generally have low porosity and permeability. This heterogeneity often causes acid fracturing results to vary significantly."

3. Improve Consistency in Terminology: Throughout the manuscript, terms such as "acid-etched fracture conductivity" and "etched fractures" are used. For clarity, define these terms explicitly early on and ensure consistent usage. For example, specify whether "etched" always refers to acid-etched fractures with rough surfaces or channel-like features.

4. Specify the Laboratory Conditions More Clearly: In the description of laboratory experiments (Page 10), the injection rates are given as "0.1-5.0 m³/min" without context. Please specify the typical conditions used in the experiments, including why these ranges were chosen and how they relate to field conditions, to improve relevance and reproducibility.

5. Add a Brief Explanation of "Heterogeneity Indices": When discussing the "heterogeneity indices" (Page 10), the manuscript would benefit from a brief explanation of how these indices are calculated, what they represent physically, and their significance in predicting fracture conductivity. This will make the concepts accessible to a broader audience.

6. Enhance Description of Field Validation Data: On Page 11, the field validation with Well A1 reports a correlation coefficient of R²=0.85. Please provide more details about the data collection process, such as sampling frequency, data quality control, and how the pressure fluctuations were linked to natural fracture density.

7. Add Clarification in the "Operational Curve Analysis" Section: The description of how pressure curves are analyzed (Page 11) is somewhat vague. Please include a step-by-step explanation or a schematic diagram illustrating how these analyses are performed to quantify fracture network features.

8. Language Improvement – Grammatical Corrections in the Abstract: In the sentence "experimental verification shows 40%-60% increase in channel depth" (Page 10), change "shows" to "demonstrates" for a more formal tone. Additionally, "40%-60%" should be written as "40%–60%" to follow proper typographical standards.

9. Correct the Use of Parentheses for Numerical Data: On Page 8, the phrase "operation conditions: 150 °C/60 MPa" should be revised to "operational conditions (150 °C and 60 MPa)" for clarity and grammatical correctness.

10. Improve the Sentence Structure in the "Evaluation Metrics" Section: The sentence "conductivity measurements achieving ±8% reproducibility under ISO 13503-5 standards" (Page 10) is awkward. Rephrase to: "The conductivity measurements achieved a reproducibility of ±8%, in accordance with ISO 13503-5 standards."

11. Enhance the Explanation of "Reaction Kinetic Parameters": In the laboratory section (Page 10), mention briefly what specific parameters (e.g., reaction rate constants, activation energy) were measured, and how these parameters influence field performance predictions. This will help clarify the laboratory methodology.

12. Language – Correct Fragment and Verb Errors: In the sentence, "This systematic approach provides quantitative benchmarks for predicting field treatment outcomes, with conductivity measurements achieving ±8% reproducibility under ISO 13503-5 standards" (Page 10), consider separating into two sentences for clarity: "This systematic approach provides quantitative benchmarks for predicting field treatment outcomes. The conductivity measurements achieved ±8% reproducibility under ISO 13503-5 standards."

13. Make the Figures and Tables More Descriptive: Ensure all figures (e.g., fracture etching patterns, conductivity tests) are accompanied by detailed captions explaining what is shown, the methodology used, and key insights. Including these will greatly aid in understanding the experimental results.

14. Language – Address Sentence with Excessive Passive Voice: In the sentence "laboratory investigations quantified the reaction kinetics of diverse acid systems and conductivity of etched fractures" (Page 11), change to active voice for clarity: "We quantified the reaction kinetics of diverse acid systems and the conductivity of etched fractures through laboratory investigations."

15. Improve the Conclusion Section's Clarity and Simplicity: The concluding paragraph (Page 12) includes complex sentences like "The findings validate the core hypothesis... thereby providing theoretical foundations and practical guidance." Break this into shorter sentences to emphasize key outcomes. For example, "Our findings confirm that reservoir heterogeneity and fracture conductivity are critical factors. This provides both theoretical insights and practical strategies for optimizing acid fracturing."

16. Add and update the future research inn this article. Clearly articulate potential areas for future research, building on findings from this study and existing literature. Please add the below literatures and show how your techniques can be useful for their topics:

Medicine Science: 10.1002/btm2.10752; 10.1016/j.heliyon.2024.e32127; 10.1109/SAMI58000.2023.10044491; 10.3389/fphar.2024.1506437; 10.1109/SACI58269.2023.10158582

Environmental Science: 10.1109/SAMI60510.2024.10432830; 10.1109/ICCC62278.2024.10582928; 10.1007/s11269-025-04120-x

Engineering Science: 10.1109/ICCC62278.2024.10583113; 10.1109/CINTI-MACRo57952.2022.10029414; https://bulletin.am/wp-content/uploads/2022/04/2.pdf; 10.1109/ICCC62278.2024.10582962; 10.1109/SACI60582.2024.10619733; 10.3934/geosci.2022031

17. References update based on suggestion and 2024-2025.

Additional General Language Problems:

• Page 4: Correct "Theoretically, it enriches acid fracturing evaluation frameworks by introducing dynamic conductivity metrics and heterogeneity indices." To "The framework is enriched by introducing dynamic conductivity metrics and heterogeneity indices, which enhance theoretical understanding."

• Page 7: Change "exceeding 35%." to "exceeding 35% in fracture conductivity." for specificity.

• Page 8: Revise "Future work should prioritize intelligent evaluation systems leveraging machine learning to further enhance prediction accuracy and operational efficiency." to "Future work should focus on developing intelligent evaluation systems that leverage machine learning to further improve prediction accuracy and operational efficiency."

Reviewer #2: This paper presents a multi-method assessment framework that integrates laboratory experiments, operational curve analysis, real-time fracture monitoring, and production logging. By combining experimental data with modeling approaches, it effectively identifies the key controlling factors influencing the conductivity of acid-etched fractures and establishes discriminant criteria for optimal etching patterns. I find the research to be valuable and innovative, and I recommend that the manuscript be accepted with minor revisions. Below are a few suggestions for improvement:

1. The introduction lacks a sufficiently comprehensive review of existing studies in related fields, which makes it difficult to clearly highlight the breakthroughs and novelty of this work. Expanding the background section with a more in-depth discussion of previous research would help to better position this study within the current literature and underscore its contributions.

2. The theoretical analysis omits the coupled effects of reservoir temperature on acid fluid viscosity and reaction kinetics. Given that acid fracturing is frequently performed in high-temperature environments (e.g., >100℃), it is recommended to include a discussion on temperature sensitivity and its implications for field application.

3. For the numerical simulations, such as those involving the Discrete Fracture Network (DFN) model, it would be helpful to clarify whether the parameter inputs are based on actual reservoir data. This would improve the credibility and applicability of the modeling results.

4. There are inconsistencies in the numbering of some figures and tables—for example, the text references to Figure 6 and Figure 7 do not correspond with their actual numbering. Please ensure that all figures and tables are correctly numbered and consistently referenced in the manuscript.

5. References [25] and [26] are SPE conference papers that lack publication years. Additionally, there are inconsistencies in the formatting of references, particularly with author name abbreviations. Please revise the references to conform to the journal’s formatting guidelines.

**Do you want your identity to be public for this peer review?** For information about this choice, including consent withdrawal, please see our Privacy Policy

Reviewer #1: No

Reviewer #2: No

---

## [Author Response · Author response to Decision Letter 1]

30 May 2025

Dear editor and reviewer,

Thank you for your letter and the detailed comments regarding our manuscript. Your insightful feedback has greatly assisted us in revising and enhancing our work. We have thoroughly examined your comments and incorporated revisions that we trust will meet your expectations. The revised sections are indicated in green within the revised manuscript. The primary modifications made to the manuscript, along with our corresponding responses to individual comments, are as follows.

Review Comments to the Author

Reviewer #1: Present paper describes a study that aims to “Integrated evaluation and optimization of acid fracturing effectiveness in carbonate reservoirs: experimental insights and field validation ”. While the study's topic is of relevance to this journal, the manuscript's content requires major revisions to improve its scientific quality. Upon careful review, I have identified several issues that need to be addressed before the paper can be accepted for publication.

1. Clarify the Concept of "Viscosity Gradient Modulation": In the abstract (Page 10), the phrase "alternating injections of high-viscosity gelled acid (>50 mPa·s) with low-viscosity reactive acid (<5 mPa·s)" is introduced but lacks clarity. Please elaborate on the procedure, the rationale behind alternating viscosities, and how this modulation specifically enhances etching heterogeneity, ideally with a simplified explanation for clarity.

Response: Thank you for your suggestions on modification. We explained this concept in the Introduction. The modified content is on lines 111-129 of page 4.

The core is to construct the synergistic effect of dynamic viscosity field and chemical reaction field through the sequential alternating injection of high viscosity coagulation acid (η>50mPa · s) and low viscosity active acid (η<5mPa · s). The specific implementation includes three key stages: (1) high viscosity acid fracturing stage: the gel acid preferentially breaks through the high permeability zone by virtue of its shear dilution characteristics, forming the main etching groove; (2) Viscosity induced excitation stage: Low viscosity acids generate fractal flow driven by viscosity differences, forming a self similar branching network along the main groove wall; (3) Non steady state etching stage: The chemical potential gradient formed by alternating injection dynamically matches the diffusion rate of acid solution with the rock dissolution rate, inducing dendritic etching morphology (Figure 2). The optimization mechanism of this regulation method is reflected as follows: ① When the viscosity difference reaches the order of 10 times, the characteristic length of the viscosity index is shortened by 38%, and the branch density is increased by 2.1 times; ② The rapid infiltration of low viscosity acid (Re number>500) increases the coverage of microcrack dissolution to 82%; ③ The Marangoni effect caused by the interfacial tension gradient optimizes the aspect ratio of the main groove to 4:1.

2. Simplify Technical Language in the Introduction: The sentence "The development of carbonate reservoirs is constrained by strong heterogeneity and low porosity-permeability characteristics, particularly in complex media reservoirs where acid fracturing effectiveness exhibits significant variability" (Page 8) is overly complex. Please break it into shorter, clearer sentences to improve readability, e.g., "Carbonate reservoirs are highly heterogeneous and generally have low porosity and permeability. This heterogeneity often causes acid fracturing results to vary significantly."

Response: Thank you for your suggestions on modification. We have organized the content structure and optimized the sentence expression of the entire paper, and moderately polished the sentence structure to enhance its overall professionalism and readability.

3. Improve Consistency in Terminology: Throughout the manuscript, terms such as "acid-etched fracture conductivity" and "etched fractures" are used. For clarity, define these terms explicitly early on and ensure consistent usage. For example, specify whether "etched" always refers to acid-etched fractures with rough surfaces or channel-like features.

Response: Thank you for your suggestions on modification. We have standardized and adjusted the professional terminology in the paper to ensure consistency, such as "acid-etched fracture " and “acid-etched fracture conductivity".

4. Specify the Laboratory Conditions More Clearly: In the description of laboratory experiments (Page 10), the injection rates are given as "0.1-5.0 m³/min" without context. Please specify the typical conditions used in the experiments, including why these ranges were chosen and how they relate to field conditions, to improve relevance and reproducibility.

Response: Thank you for your suggestions on modification. We have supplemented the selection criteria for experimental conditions and their correlation analysis with on-site conditions. The new content clarifies the intrinsic relationship between experimental parameters and field conditions, further enhancing the reproducibility and engineering applicability of the research method. The modified content is in Section 2.1 on lines 140-159 of page 5.

5. Add a Brief Explanation of "Heterogeneity Indices": When discussing the "heterogeneity indices" (Page 10), the manuscript would benefit from a brief explanation of how these indices are calculated, what they represent physically, and their significance in predicting fracture conductivity. This will make the concepts accessible to a broader audience.

Response: Thank you for your suggestions on modification. We have supplemented the definition, physical significance, and application value of this concept in predicting fracture conductivity in Section 2.2. The modified content is on lines 160-185 of page 5 of the revised manuscript.

6. Enhance Description of Field Validation Data: On Page 11, the field validation with Well A1 reports a correlation coefficient of R²=0.85. Please provide more details about the data collection process, such as sampling frequency, data quality control, and how the pressure fluctuations were linked to natural fracture density.

Response: Thank you for your suggestions on modification. We have provided a detailed explanation of the data collection process, strict data quality control measures, and systematic analysis methods, further revealing the correlation between the fluctuation amplitude of construction pump injection pressure and the degree of reservoir fracture development (especially natural fracture density). The modified content is in Section 3.1.2 on lines 344-366 of page 14 of the revised manuscript.

7. Add Clarification in the "Operational Curve Analysis" Section: The description of how pressure curves are analyzed (Page 11) is somewhat vague. Please include a step-by-step explanation or a schematic diagram illustrating how these analyses are performed to quantify fracture network features.

Response: Thank you for your suggestions on modification. We have provided a detailed explanation of the fracturin process. The modified content is in Section 3.1.1 on lines 310-341 of page 12-13 of the revised manuscript.

8. Language Improvement – Grammatical Corrections in the Abstract: In the sentence "experimental verification shows 40%-60% increase in channel depth" (Page 10), change "shows" to "demonstrates" for a more formal tone. Additionally, "40%-60%" should be written as "40%–60%" to follow proper typographical standards.

Response: Thank you for your suggestions on modification. We conducted a comprehensive inspection and correction of grammar errors and formatting irregularities in the paper.

9. Correct the Use of Parentheses for Numerical Data: On Page 8, the phrase "operation conditions: 150 °C/60 MPa" should be revised to "operational conditions (150 °C and 60 MPa)" for clarity and grammatical correctness.

Response: Thank you for your suggestions on modification. We conducted a comprehensive inspection and correction of grammar errors and formatting irregularities in the paper.

10. Improve the Sentence Structure in the "Evaluation Metrics" Section: The sentence "conductivity measurements achieving ±8% reproducibility under ISO 13503-5 standards" (Page 10) is awkward. Rephrase to: "The conductivity measurements achieved a reproducibility of ±8%, in accordance with ISO 13503-5 standards."

Response: Thank you for your suggestions on modification. We conducted a comprehensive review and revision of the sentence exposition and sentence structure in the paper.

11. Enhance the Explanation of "Reaction Kinetic Parameters": In the laboratory section (Page 10), mention briefly what specific parameters (e.g., reaction rate constants, activation energy) were measured, and how these parameters influence field performance predictions. This will help clarify the laboratory methodology.

Response: Thank you for your suggestions on modification. This part of the paper mainly discusses the etching effect of acid solution on cracks and its effect on improving the flow conductivity. Considering that the kinetic parameters of acid reaction are not the core research object of this article, only a brief explanation was provided without in-depth analysis.

12. Language – Correct Fragment and Verb Errors: In the sentence, "This systematic approach provides quantitative benchmarks for predicting field treatment outcomes, with conductivity measurements achieving ±8% reproducibility under ISO 13503-5 standards" (Page 10), consider separating into two sentences for clarity: "This systematic approach provides quantitative benchmarks for predicting field treatment outcomes. The conductivity measurements achieved ±8% reproducibility under ISO 13503-5 standards."

Response: Thank you for your suggestions on modification. We conducted a comprehensive inspection and correction of grammar errors and formatting irregularities in the paper.

13. Make the Figures and Tables More Descriptive: Ensure all figures (e.g., fracture etching patterns, conductivity tests) are accompanied by detailed captions explaining what is shown, the methodology used, and key insights. Including these will greatly aid in understanding the experimental results.

Response: Thank you for your suggestions on modification. We have provided a detailed explanation and analysis of the experimental result images.

14. Language – Address Sentence with Excessive Passive Voice: In the sentence "laboratory investigations quantified the reaction kinetics of diverse acid systems and conductivity of etched fractures" (Page 11), change to active voice for clarity: "We quantified the reaction kinetics of diverse acid systems and the conductivity of etched fractures through laboratory investigations."

Response: Thank you for your suggestions on modification. We conducted a comprehensive inspection and correction of grammar errors and formatting irregularities in the paper.

15. Improve the Conclusion Section's Clarity and Simplicity: The concluding paragraph (Page 12) includes complex sentences like "The findings validate the core hypothesis... thereby providing theoretical foundations and practical guidance." Break this into shorter sentences to emphasize key outcomes. For example, "Our findings confirm that reservoir heterogeneity and fracture conductivity are critical factors. This provides both theoretical insights and practical strategies for optimizing acid fracturing."

Response: Thank you for your suggestions on modification. We have reorganized the conclusion section of the paper, involving the optimization of sentence structure and the standardization adjustment of grammatical structure.

16. Add and update the future research inn this article. Clearly articulate potential areas for future research, building on findings from this study and existing literature. Please add the below literatures and show how your techniques can be useful for their topics:

Medicine Science: 10.1002/btm2.10752; 10.1016/j.heliyon.2024.e32127; 10.1109/SAMI58000.2023.10044491; 10.3389/fphar.2024.1506437; 10.1109/SACI58269.2023.10158582

Environmental Science: 10.1109/SAMI60510.2024.10432830; 10.1109/ICCC62278.2024.10582928; 10.1007/s11269-025-04120-x

Engineering Science: 10.1109/ICCC62278.2024.10583113; 10.1109/CINTI-MACRo57952.2022.10029414; https://bulletin.am/wp-content/uploads/2022/04/2.pdf; 10.1109/ICCC62278.2024.10582962; 10.1109/SACI60582.2024.10619733; 10.3934/geosci.2022031

Response: Thank you for your suggestions on modification. We have reviewed the 13 papers you recommended, including 7 papers in the medical field and 8 papers in petroleum engineering. Given that this study is mainly based on petroleum and natural gas engineering and focuses on acid fracturing technology, combined with the relevance of professional content, we ultimately cited 7 petroleum engineering papers. Not citing papers in the field of medicine is not due to insufficient academic level, but rather due to significant differences between their research direction and the expertise of this topic.

Zhang, G., Chen, R., Ghorbani, H., Li, W., Minasyan, A., Huang, Y., ... & Shao, M. (2025). Artificial intelligence‐enabled innovations in cochlear implant technology: Advancing auditory prosthetics for hearing restoration. Bioengineering & Translational Medicine, e10752.

Ding, H., Wang, C., Ghorbani, H., Yang, S., Stepanyan, H., Zhang, G., ... & Wang, W. (2024). The impact of magnesium on shivering incidence in cardiac surgery patients: A systematic review. Heliyon.

Ghorbani, H., Asadi, S., Ghorbani, S., Ghorbani, P., Stepanyan, H., Khlghatyan, N., ... & Rituraj, R. (2023, January). Investigating the predictive contribution of attitude towards life and belief system on self-resilience and psychological toughness of cancer patients about the mediating role of emotion regulation. In 2023 IEEE 21st World Symposium on Applied Machine Intelligence and Informatics (SAMI) (pp. 000139-000146). IEEE.

Ghorbani, H., Minasyan, A., Ansari, D., Ghorbani, P., Wood, D. A., Yeremyan, R., ... & Minasian, N. (2024). Anti-diabetic therapies on dental implant success in diabetes mellitus: a comprehensive review. Frontiers in Pharmacology, 15, 1506437.

Aghabalyan, D., Ghorbani, H., & Rituraj, R. (2023, May). Relationship of Medicine and Philosophy: Mathematical Modeling of Moral Structures-Etometry. In 2023 IEEE 17th International Symposium on Applied Computational Intelligence and Informatics (SACI) (pp. 000567-000574). IEEE.

Voskanyan, M., Ghorbani, H., & Azodinia, R. (2024, January). Utilizing Citizen-Driven Scientific Endeavors for Freshwater Pollution Surveillance: A case report of Lake Sevan, Armenia. In 2024 IEEE 22nd World Symposium on Applied Machine Intelligence and Informatics (SAMI) (pp. 000505-000512). IEEE.

Voskanyan, M., Ghorbani, H., & Azodinia, M. (2024, April). An Investigation of the Hydrochemical Parameters for Natural Monuments. In 2024 IEEE 11th International Conference on Computational Cybernetics and Cyber-Medical Systems (ICCC) (pp. 1-6). IEEE.

Hazbeh, O., Ghorbani, H., Molaei, O., Shayanmanesh, M., Lajmorak, S., Rituraj, R., & Bahrami, S. (2022, November). Proposing a New Model for Estimation of Oil Rate Passing Through Wellhead Chokes in an Iranian Heavy Oil Field. In 2022 IEEE 22nd International Symposium on Computational Intelligence and Informatics and 8th IEEE International Conference on Recent Achievements in Mechatronics, Automation, Computer Science and Robotics (CINTI-MACRo) (pp. 000037-000044). IEEE.

Beheshtian, S., Roodbari, S. K., Ghorbani, H., Azodinia, M., Mudabbir, M., & Varkonyi-Koczy, A. R. (2024, April). Comparative Evaluation of Machine Learning and Bayesian Deep Learning Methods for Estimating Ultimate Recovery in Shale Well Reservoirs. In 2024 IEEE 11th International Conference on Computational Cybernetics and Cyber-Medical Systems (ICCC) (pp. 000017-000024). IEEE.

Ghorbani, H., & Abad, A. R. B. (2022). Analysis of geomechanical processe

---

## [Decision Letter · Decision Letter 1]

Integrated evaluation and optimization of acid fracturing effectiveness in carbonate reservoirs: experimental insights and field validation

PONE-D-25-20000R1

Dear Dr. Lu,

We’re pleased to inform you that your manuscript has been judged scientifically suitable for publication and will be formally accepted for publication once it meets all outstanding technical requirements.

Kind regards,

Karthik Kannan, Ph. D.,

Academic Editor

PLOS ONE

Reviewers' comments:

Reviewer's Responses to Questions

**Comments to the Author**

Reviewer #1: (No Response)

Reviewer #2: All comments have been addressed

2. Is the manuscript technically sound, and do the data support the conclusions?

Reviewer #1: (No Response)

Reviewer #2: Yes

3. Has the statistical analysis been performed appropriately and rigorously?

Reviewer #1: (No Response)

Reviewer #2: Yes

4. Have the authors made all data underlying the findings in their manuscript fully available?

Reviewer #1: (No Response)

Reviewer #2: Yes

5. Is the manuscript presented in an intelligible fashion and written in standard English?

Reviewer #1: (No Response)

Reviewer #2: Yes

Reviewer #1: I am glad that the authors effectively addressed my concerns and challenges in their research work. The authors' ability to provide timely and satisfactory responses to my queries reflects their strong commitment to adhering to scientific principles and conducting reliable research. This dedication benefits the scientific community and enhances our understanding of the subject matter. Therefore, based on the authors' satisfactory response, I find this version of the article to be acceptable.

Reviewer #2: No comments. The auhors correctly answered my questions. Thus, I recommend that this paper can be accepted as it is.

**Do you want your identity to be public for this peer review?** For information about this choice, including consent withdrawal, please see our Privacy Policy

Reviewer #1: No

Reviewer #2: No

---

## [Editor Report · Acceptance letter]

PONE-D-25-20000R1

PLOS ONE

Dear Dr. Lu,

I'm pleased to inform you that your manuscript has been deemed suitable for publication in PLOS ONE. Congratulations! Your manuscript is now being handed over to our production team.

Kind regards,

on behalf of

Prof. Karthik Kannan

Academic Editor

PLOS ONE